# Iron and Copper Doped Zinc Oxide Nanopowders as a Sensitizer of Industrial Energetic Materials to Visible Laser Radiation

**DOI:** 10.3390/nano12234176

**Published:** 2022-11-24

**Authors:** Anton S. Zverev, Natalya N. Ilyakova, Denis R. Nurmukhametov, Yulia N. Dudnikova, Dmitry M. Russakov, Valery M. Pugachev, Anatoly Y. Mitrofanov

**Affiliations:** 1The Federal Research Center for Coal and Coal Chemistry, Institute of Coal Chemistry and Chemical Materials Science, Siberian Branch, Russian Academy of Sciences, 650000 Kemerovo, Russia; 2Institute of Fundamental Science, Kemerovo State University, 650000 Kemerovo, Russia

**Keywords:** composite nanopowder, zinc oxide, combustion synthesis, laser initiation, photochemistry, photocatalysis, pentaerythritol tetranitrate, diffuse reflection spectroscopy, energetic materials

## Abstract

The development of methods ensuring reliable control over explosive chemical reactions is a critical task for the safe and efficient application of energetic materials. Triggering the explosion by laser radiation is one of the promising methods. In this work, we demonstrate a technique of applying the common industrial high explosive pentaerythritol tetranitrate (PETN) as a photosensitive energetic material by adding zinc oxide nanopowders doped with copper and iron. Nanopowders of ZnO:Fe and ZnO:Cu able to absorb visible light were synthesized. The addition of one mass percent nanopowders in PETN decreased the threshold energy density of its initiation through Nd:YAG laser second harmonic (2.33 eV) by more than five times. The obtained energetic composites can be reliably initiated by a CW blue laser diode with a wavelength of 450 nm and power of 21 W. The low threshold initiation energy and short irradiation exposure of the PETN-ZnO:Cu composite makes it applicable in laser initiation devices. PETN-ZnO:Cu also can be initiated by an infrared laser diode with a wavelength of 808 nm. The proposed photochemical mechanism of the laser-induced triggering of the explosion reaction in the studied energetic composites was formulated. The results demonstrate the high promise of using nanomaterials based on zinc oxide as a sensitizer of industrial energetic materials to visible laser radiation.

## 1. Introduction

The initiation of combustion or explosion of energetic materials (EM) is a crucial stage for many technological processes and device operations. Laser radiation is a promising tool for making EM initiation safer and more controllable [1,2,3,4,5]. Laser initiation has several advantages, such as electromagnetic noise immunity, better control of the ignition process, and lower weight of fiber-optic systems relative to wire electro-initiation systems. But the critical advantage of the approach is a potential opportunity to directly initiate combustion or explosion of the relatively insensitive energy materials, for example, direct initiation of high explosives (HE) [1,6,7,8]. The main technological problem is the weak interaction of laser radiation with EM. The most common commercially available industrial lasers emit in the visible and near-infrared (NIR) regions, and most industrial EMs are transparent in these spectral regions [9].

To initiate an explosive reaction, it is necessary to overcome the potential barrier by transferring additional energy to the EM. Fast heating or mechanical impulse are typical techniques for such energy transfer. From this point of view, the efficiency of laser initiation directly depends on the optical absorbance of the material, which can be increased by introducing an opaque additive (carbon and metal micro- or nanoparticles) [6,10,11,12,13,14]. This approach ignores a fundamental property of electromagnetic radiation: i.e., its ability to excite the electron system of material. The chemical properties of the material can significantly differ in the initial and excited states. A suitable photocatalyst irradiated by ultraviolet or visible light can drive processes that are inefficient or impossible under other conditions. Photocatalytic chemical reactions are widely applied to solve various technical tasks in various fields such as modern green energy, water purification, organic synthesis, and medicine [15,16,17,18,19]. The role of photochemical processes in the laser initiation of an EM has been discussed since the first publications on this topic [20] and is still relevant today [21,22,23,24,25]. The key property of radiation is its ability to transfer the energetic material molecule to an excited state from which the decomposition process can occur with no or a lower potential barrier. In this case, an effective sensitizing additive should act as a mediator in the transfer of excitation to the EM but should not increase the absorption index of the energy composite to high values. Specific compounds [21,23,25] with a particularised mechanism of action and simple inorganic materials without special treatment [24,26] can be applied as photosensitizers.

This research aims to control pentaerythritol tetranitrate (PETN) sensitivity to visible laser radiation by the addition of nanostructured zinc oxide powders doped with iron (ZnO:Fe) and copper (ZnO:Cu). Photoinduced charge transfer at the photosensitizer–PETN interface can create a metastable PETN anion-radical, which can spontaneously (or with a negligible potential barrier) decompose and trigger an explosive reaction [27]. This process was demonstrated by us experimentally for the PETN-MgO composite initiated by ultraviolet laser radiation with a wavelength of 355 nm [24]. Starting this work, we assumed that zinc oxide nanosized heterostructures could demonstrate a similar photocatalytic effect and could be a promising additive for increasing PETN sensitivity to visible laser radiation.

Doping zinc oxide with iron and copper can lead to the formation of impurity centers [28,29] or phases of iron oxides [30], zinc ferrite ZnFe_2_O_4_ [31], and copper (II) oxide CuO [32,33]. An impurity center creates an acceptor or donor local state in the ZnO band gap, which allows for the generation of band charges by radiation with lower quantum energy. In the case of two-phase material, a specific position of the Fermi levels of semiconductors can create a Type-II heterojunction scheme, where edges of valence and conduction bands of one phase are higher than another. This creates the opportunity to localize photoexcited electrons and holes in two different phases. These separated charges can be effectively involved in redox reactions. Such doped nanomaterials and heterostructures based on zinc oxide are effectively used as photocatalysts for ecological [28,29,30,31] and sustainable energy [32,33] purposes. In this work, we considered the possibility of using such materials to produce a light-sensitive energetic composite.

## 2. Materials and Methods

### 2.1. Materials Synthesis and Characterisation

The combustion method [34,35] was used for the synthesis of ZnO, ZnO:Fe, and ZnO:Cu nanopowders. A small amount of distilled water and 4 g of zinc oxide powder (99.0%, JSC LenReactiv, Saint-Petersburg, Russia) were placed in a 250 mL beaker. ZnO was completely dissolved in concentrated nitric acid (56%, JSC LenReactiv, Saint-Petersburg, Russia). The solution was diluted with distilled water to 40 mL. Iron (II) acetate tetrahydrate was used for ZnO:Fe preparation, and copper (II) nitrate (99.7%, JSC LenReactiv, Saint-Petersburg, Russia) was used for ZnO:Cu. Molar proportions of both metals relative to zinc were 5% in the resulting material. A portion of 3.4 g of sucrose was added to the solution. The resulting solution was brought to boil and evaporated to a volume of about 20 mL. A beaker of the solution was placed in a muffle furnace preheated to a temperature of 250 °C and kept there for 30 min. The resulting product was placed in a ceramic bowl and calcinated in a muffle furnace for 3 h at a temperature of 450 °C for carbon residue annealing. The resulting material was a brittle and friable ceramic foam. The materials were ground in a hand agate mortar into fine homogeneous powders.

A Jeol JEM-2100 microscope (JEOL Ltd., Tokyo, Japan) was used to obtain transmission electron microscopy images. Powder diffractograms were registered using the DRON-8 diffractometer (Bourevestnik, JSC, St. Petersburg, Russia) with an X-ray wavelength of 1.5406 Å.

The ASAP-2020 (Micromeritics Instrument Corporation, Norcross, GA, USA) high-performance adsorption analyzer for measuring surface area was used for the study of the nanopowders’ surface characteristics. The specific surface area (S_BET_, m^2^/g) of the samples was calculated from the N_2_ adsorption-desorption isotherms at −196 °C (77 K). Before these adsorption measurements were taken, the test samples were dried in an oven at a temperature of 105 ± 5 °C to constant weight and then evacuated directly in a port of the analyzer at 110 °C for 12 h to a residual pressure of no more than 0,67 mPa. Nitrogen adsorption-desorption isotherms were measured in the range of equilibrium-relative vapor pressures from 10^−3^ to 0.995 P/P_0_. The specific pore surface area was calculated using the Brunauer-Emmett-Taylor (BET) method. The micropore volume was determined using the comparative t-Plot method. The mesopore volume was determined using the Barret-Joyner-Halenda (BJH) method. The average pore diameter was estimated by the formula D_pores_ = 4 V_Σ_/S according to the BET method. The mesopore volume was calculated from the size distribution of mesopores (BJH method). The measurement error was 5–7%.

### 2.2. Diffuse Reflection Spectroscopy

Diffuse reflectance spectra of ZnO, ZnO:Fe, and ZnO:Cu were measured by a Shimadzu UV-3600 scanning spectrophotometer (Shimadzu Corporation, Kyoto, Japan) equipped with a UV-VIS-NIR integrating sphere attachment ISR-3100. The obtained diffuse-reflection spectra were transformed using the Kubelka-Munk formula: F(R) = (1 − R)^2^/2 R, where R is a reflection percentage. The reflectance spectrum of the obtained zinc oxide nanopowder with deposed PETN was also measured. The zinc oxide was mixed with PETN by grinding in an agate mortar. The concentration of PETN in the mixture was 3 mass percent. The mixture was heated in a drying cabinet for 10 min at 143 °C, which is a little above the melting point of PETN, to achieve a better coating of ZnO particles by energetic material. Diffuse reflectance spectrum of obtained composite ZnO-PETN was measured by the same procedure.

### 2.3. Photoacoustic Measurements

The effective extinction index was determined by the photoacoustic method detailed in [36]. The schematic diagram of measurement is shown in Figure 1. An acoustic wave induced by second harmonic radiation (532 nm) of the LQ929 (SOLAR Laser Systems, Minsk, Belarus) Nd:YAGlaser was measured using a piezo-sensor and recorded by the LeCroy WJ332A oscilloscope. The front-increasing site of the obtained oscillogram (Figure 1) was approximated by the exponential dependence according to formula 1. If τ is significantly higher than the laser pulse duration (14 ns), it is possible to evaluate the effective extinction index (k_eff_) of the sample.
U(t < 0)~exp(t/τ),(1)
where τ = (k_eff_c)^−1^ and c—is the sound speed of PETN (2500 m/s).

### 2.4. Laser Initiation

PETN and ZnO:Fe or ZnO:Cu powders were placed into the ceramic beaker and sonicated in 10 mL of pure hexane for 10 min. The concentration of the sensitizer additive was 1 wt%. After the evaporation of hexane, the obtained mixtures were used for the preparation of pressed tablets. The pressing was carried out under the pressure of 0.2 MPa. The resulting sample was a tablet of 3 mm in diameter, 0.5 mm thickness, and 7 mg mass, pressed into an aperture of the stainless-steel holder disc.

The obtained samples were initiated by laser radiation of Nd:YAG laser LDLP10-1500 (LaserDream Optronics Co.,Ltd., Beijing, China) with a pulse duration of 14 ns. The second harmonic (532 nm) was used (Appendix A). Diode lasers AP-OEM-YD-P-0450-20-5A1K (Changchun New Industries Optoelectronics Technology Co., Ltd., Changchun, China) with a wavelength of 450 nm and Xinland XL808H10W (Xi’an Xinland International Co., Ltd., Xi’an, China) with a wavelength of 808 nm were used as a CW laser source. The energy density was monitored by a PE50BF-DIF-C pyroelectric energy sensor (Ophir Optronics, Jerusalem, Israel) and a Gentec QE25LP-H-MB-QED-INT-D0 energy sensor (Gentec Electro-Optics Inc., Quebec City, QC, Canada). The part of the laser beam passing through the empty sample holder plate, installed in the same place as during the initiation test, was measured for a determination of the energy density. The beam of a CW diode laser with a wavelength of 450 nm had a pronounced inhomogeneous power distribution over the cross-section, which was close to the Gaussian. The energy density was estimated using the central part of the beam with a diameter of 1.2 mm (Appendix A). A beam of the 808 nm laser diode was focused by the lens. It allowed us to achieve the power density necessary for sample initiation. The resulting cross-section had the shape of a rectangle with sides of 3 mm and 0.09 mm. The unfocused laser beam print of the IR diode and its energy distribution is shown in Appendix A. The schematic diagram of the experimental setup is shown in Figure 1.

A successful initiation result included an intense acoustic response, the absence of sample particles in both the holder disk and the explosion chamber, and the imprint left by the explosion products on the base surface. In the case of CW laser initiation, the dependence of the percentage of successfully initiated samples before the specific moment of irradiation time (energy density) from energy density was plotted (Figure 1). The experimental dots were approximated by the relationship between the probability of the sample explosion and the energy density of the laser radiation proposed in [37]. The laser initiation thresholds were determined as the energy density required for the successful initiation of 50% of the samples.

## 3. Results

### 3.1. Phase Composition, Morphology and Optical Properties of ZnO Based Nanopowders

TEM images of ZnO, ZnO:Fe, and ZnO:Cu nanosized powder are shown in Figure 2. All materials were polycrystalline agglomerates of nanoparticles. ZnO:Cu consisted of relatively large particles covered with much smaller ones. The size distributions of the particles observed on TEM images are also shown in Figure 2.

Nanopowder diffractograms and XRD patterns for the proposed crystalline phases of the materials are shown in Figure 3. The diffraction pattern of the ZnO:Cu nanopowder contained the patterns of copper (II) oxide [38] in addition to the patterns of zinc oxide [39]. There were no visible maxima other than the zinc oxide phase on the diffractogram of the ZnO:Fe powder. This indicated the incorporation of iron into the crystal lattice as impurity atoms. However, we assumed that the most probable phase of zinc ferrite (ZnFe_2_O_4_) [40] was formed from the added iron. In that case, its concentration in the composite was 2.7 mol% or less. The most intensive XRD patterns of this phase were located in the region of the most intense patterns of zinc oxide, which made it difficult to distinguish additional phases against the background of ZnO patterns. The diffractograms of nanosized heterostructures had no significant differences in the position of the zinc oxide patterns maxima both from each other and from pure zinc oxide. The reflections of ZnO:Fe had a noticeably narrower width at half maximum. We estimated the average sizes of crystallites in nanopowders using the Scherrer equation. It was 30 nm for ZnO patterns in ZnO:Fe nanopowder. In the case of ZnO:Cu, the average crystallite sizes of zinc oxide and copper oxides were 100 nm and 50 nm, respectively. The average particle size of pure zinc oxide nanopowder obtained by the same method was about 70 nm.

Joint analysis of the XRD and TEM results allowed us to conclude that ZnO:Fe was represented mainly by 20–30 nm nanocrystallites with a small amount of large-size particles (about 70–100 nm). The ZnO and ZnO:Cu nanopowders had a significantly wider particle size distribution and higher concentration of larger crystallites from several tens to 100 nm, which were more visible on the diffractograms.

The surface textural characteristics of the obtained powders as the values of the specific surface (S_BET_, m^2^/g), pore volumes (a total pore volume—V_Σ_, cm^3^/g; a volume of micro- and mesopores V_micro_, V_meso_, cm^3^/g), and average pore diameter (D_pores_, Å) of the studied samples are shown in Table 1. The ZnO:Fe nanopowder had the largest specific surface area, while ZnO:Cu and ZnO had a specific surface area of approximately three and two times less, respectively. These values are in good agreement with the results of the above study of the materials’ morphology. In all materials, the porous structure was predominantly represented by mesopores.

Figure 4a shows the diffuse reflectance spectrum of ZnO, ZnO:Cu, and ZnO:Fe nanopowders. The edge of the intrinsic absorption of zinc oxide was observed at 3.2–3.3 eV, which is in good agreement with its band gap [41] at room temperature. The ZnO intrinsic absorption edge spectrum region had a flatter slope for both ZnO:Cu and ZnO:Fe. The band gaps of materials were determined by approximating the intrinsic absorption edge region of zinc oxide in hv-(F(R)hv)^2^ coordinates (Figure 4b). For ZnO, ZnO:Cu, and ZnO:Fe, they were 3.29, 3.22, and 3.215 eV, respectively. The lower optically determined bad gap was shown for ZnO-ZnFe_2_O_4_ [31] and ZnO-CuO [32,33] nanocomposites, which could be an additional marker for the correctness of the above-determined phase composition of the materials. ZnO:Fe and ZnO:Cu powders had a noticeable absorption in the visible region. ZnO:Cu had a noticeable optical absorption in the full-length visible spectral region of up to 1.46 eV (850 nm). The spectrum showed two local maxima at about 2.7 and 2 eV. The ZnO:Fe visible region of the spectrum was represented by a monotonically decreasing wing, on which a weak local absorption maximum of 2.6 eV could be distinguished. The spectrum form was practically identical to the spectrum of ZnFe_2_O_4_ measured in [31]. It allowed us to assume the containment of the zinc ferrite in ZnO:Fe. The absorption in the visible region was about two orders of magnitude lower than in the intrinsic absorption region for both materials.

The spectrum of ZnO nanopowder covered by PETN was also studied (Figure 4a). An absorption shoulder outside the intrinsic absorption region of zinc oxide was found. A similar effect was observed for PETN deposited on corundum [26] and magnesium oxide [24]. It was explained by the photoinduced transfer of oxide valence electrons to unoccupied PETN orbitals. In the case of dielectric materials, the lowest vacant states of PETN (LUMO) were located in the band gap of the oxide, but in the case of a narrower gap ZnO they should be located near the bottom of the conduction band. The results of the spectroscopic study did not allow us to consider pure ZnO nanopowder as an efficient photosensitizer due to its low absorption, but they did allow us to formulate several proposed mechanisms for charge transfer on PETN (Figure 4c) for copper- and iron-doped nanopowders. The formation of PETN anion-radical by the charge transfer can be considered a potential trigger for the explosive decomposition reaction [27].

### 3.2. Laser Initiation Tests

The explosion probabilities of pure PETN [24] and its composites with the addition of ZnO:Fe and ZnO:Cu powders as a function of the energy density of the second harmonic of the Nd:YAG laser are presented in Figure 5a. The initiation threshold of pure PETN corresponds to ≈3.3 J/cm^2^, while a 100% probability of explosion was not achieved even at 4 J/cm^2^. Composites containing 1 wt% of ZnO:Fe and ZnO:Cu nanopowders demonstrate significantly higher sensitivity to the second harmonic of the Nd:YAG laser. The initiation threshold of both composites was 0.7 J/cm^2^. Identical photosensitization efficiency of nanopowders, which obviously differ by the absorption index on laser wavelength, can be explained by the significantly higher fineness and specific surface area of ZnO:Fe.

Figure 5b shows the values of the initiation thresholds of PETN-ZnO:Cu and ZnO:Fe, as well as PETN-based composites with light-absorbing additives of metal nanoparticles [6,10,42,43,44]. The materials containing aluminum nanoparticles have the highest laser sensitivity due to nano-aluminum’s high chemical activity [4,45]. The initiation thresholds of composites with the addition of ZnO:Fe and ZnO:Cu are not inferior to most of the above additives. Effective extinction indexes at the initiation wavelength were compared to understand the contributions of the thermal and photochemical initiation mechanisms. The effective extinction index on Nd:YAG laser second harmonic (2.33 eV) of the PETN-ZnO:Cu and PETN-ZnO:Fe was measured by the photoacoustic method. The effective extinction index of the PETN-ZnO:Cu composite was 85–90 cm^−1^. The PETN-ZnO:Fe samples were so transparent that the effective extinction index could not be measured. An acoustic signal formed by a composite layer heated by laser radiation was less intensive than the signal produced by acoustic delay. The PETN-ZnO:Fe effective extinction index should be lower than 20 cm^−1^. The effective extinction index of PETN composites with the addition of aluminum, iron, and nickel nanopowder was about 100–300 cm^−1^ [36,43,44]. The measured extinction index of the PETN-ZnO:Cu composite was about 2–3 times lower, and the measured extinction index of PETN-ZnO:Fe was at least an order of magnitude lower than energetic composites with metal nanoparticles.

The absorption index measured by the photoacoustic method characterized the absorbed radiation inducing the material heating and, therefore, can be considered a criterion of the material heating efficiency. If we suggest the purely thermal mechanism of initiation of composites, we must conclude that nanopowders based on zinc oxide should be noticeably less effective sensitizing additives than metal nanopowders. However, ZnO:Cu and ZnO:Fe showed comparable or superior results. This comparison allowed us to assume a significant contribution of the photochemical mechanism in the initiation process of PETN-ZnO:Cu and PETN-ZnO:Fe composites via the second harmonic of the Nd:YAG laser. Previously studied photochemical sensitizing additives, such as 9,10-phenanthrenequinone and magnesium oxide powder, also demonstrated a markedly lower efficiency in the initiation threshold [23,24].

The dependencies of the explosion probability of PETN-ZnO:Fe and PETN-ZnO:Cu composites from the energy density of a CW diode laser with a wavelength of 450 nm are shown in Figure 5a. Pure PETN cannot be initiated under these conditions. The initiation threshold of PETN-ZnO:Cu was three times lower than PETN-ZnO:Fe. Threshold energy densities were equal to 26 and 95 J/cm^2^. These values seem to be extremely high compared to the threshold energy density of pulsed laser tests, which is less than 1 J/cm^2^. However, this difference is not so dramatic if we look at the threshold jointly with the power densities of the lasers. The initiating radiation power density of the laser diode was 450 W/cm^2^, and that of the Nd:YAG laser was 50 MW/cm^2^. The threshold ignition energy density increased only a factor of a few tens, while the laser radiation power density decreased by five orders of magnitude. Therefore, nanopowders more efficiently sensitize PETN even to CW than to pulsed laser radiation.

A similar dependence for PETN-ZnO:Cu samples initiated by a CW laser diode with a wavelength of 808 nm is shown in Figure 4a. A composite with the addition of ZnO:Fe and pure PETN were not initiated under these conditions. Successfully initiation of all PETN-ZnO:Cu samples was not achieved either.

PETN is an extremely interesting model energetic material for ignition studies using low-power CW lasers due to its relatively low melting point, which creates a critical problem. In the case of initiation of PETN composites with metal nanoparticles, the destruction of the sample caused by melting and thermal decomposition is a more probable outcome than the explosion of the sample. The melting of the sample, accompanied by combustion without an explosion, in the case of the PETN doped by the hollow gold nanospheres initiated by the 808 nm laser diode was reported in [7]. At the same time, similar composites based on noticeably less sensitive Keto-RDX and HMX were successfully initiated. The authors consider the low melting temperature of PETN and the relatively big difference between its melting and ignition temperatures as the main reasons for this result. The explosion of more than half of the tested PETN-ZnO:Cu samples was successfully initiated under similar conditions. The unsuccessfully initiated samples did not have signs of intense melting or combustion of the material. A cloudy spot covered with cracks and cavities, coinciding with the irradiated region, was observed on the surface of the samples (Figure 6a–c). It indicates that the sample did not heat up to the melting point of PETN (141.3 °C) even at exposure to 4200 J/cm^2^, but instead partially decomposed.

The nanopowders synthesized in this study, especially ZnO:Cu, provided not only relatively low initiation thresholds but also the guaranteed initiation of samples by a blue laser diode, when the combustion without a complete explosion of the sample was not observed. It is worth noting that most studies devoted to both the photosensitization of industrial energetic materials and the development of new materials for laser initiation, with some exceptions [7,14,22,46], consider the self-sustaining combustion of the sample as a criterion for successful ignition [11,12,47]. In this work, a more intensive process was observed as a successful initiation, despite the small size of the sample. The experimental conditions do not allow us to determine this process as a detonation, nevertheless, the high intensity of the process was indicated by the acoustic response, as well as the external consequences of the explosion. Photographs of a piece of aluminum foil placed between the base and the sample, taken after successful initiation, are shown in Figure 6d. Traces on the base surface indicate that the explosion of the 0.5 mm thickness and 3 mm diameter tablet created pressure sufficient to transfer the aluminum to a liquid state.

The samples’ high sensitivity to visible laser radiation, relatively low absorption index, and absence of noticeable signs of intense heating allow us to assume the significant contribution of photochemical mechanisms to the initiation process. If we consider the charge transfer as a key stage of the photochemical laser initiation [27], then we can formulate a possible mechanism of photochemical initiation for studied PETN-based composites. Electrons can be excited to the zinc oxide conduction band either by transfer from the conduction band of copper oxide or iron-containing phase, for example, zinc ferrite, or with the participation of impurity defects. The holes formed in the valence band in zinc oxide can transfer to the valence band of the addition phase, which reduces the probability of charge-carrier recombination in ZnO. This scheme can promote the efficient generation of electrons in the conduction band of zinc oxide and their transfer to the PETN molecules, leading to the formation of metastable anion radicals.

## 4. Conclusions

Nanopowders of pure zinc oxide and zinc oxides doped with copper and iron were synthesized by the combustion method. The ZnO:Cu was a composite of phases of zinc and copper (II) oxides. We could not determine any crystalline phases except zincite in ZnO:Fe, but the optical absorption spectrum point contained zinc ferrite. ZnO:Fe had the highest fineness and specific surface area among all obtained nanomaterials.

ZnO:Fe absorbed radiation in the visible region up to 2.1 eV, and ZnO:Cu up to the short-wavelength edge of the IR region (1.45 eV). An analysis of the spectra suggested the possibility of generating electrons in the conduction band of zinc oxide via visible laser radiation due to the photogeneration of charges in narrower gap phases (CuO, ZnFe_2_O_4_) followed by electron transfer to ZnO or generation of electrons through the impurity defects of the zinc oxide. The long-wavelength absorption region in the diffuse reflectance spectra of ZnO coated by PETN indicated the location of PETN vacant states near the ZnO conduction band bottom, which in turn indicated the possibility of charge transfer at the ZnO-PETN interface. This proposed photochemical mechanism could be a tool for triggering the PETN explosion reaction.

The sensitivity of PETN composites with 1 wt.% of ZnO:Fe and ZnO:Cu to the second harmonic of pulsed Nd:YAGlaser was not noticeably inferior to the sensitivity of energetic composites with the addition of metal nanoparticles, which have a much higher effective absorption index. The laser initiation threshold of both PETN-ZnO:Cu and PETN-ZnO:Fe was 0.7 J/cm^2^.

The addition of 1 wt.% of ZnO:Fe and ZnO:Cu made possible the initiation of PETN using a laser diode with a wavelength of 450 nm and a power of 21 W. The initiation threshold energy density of PETN-ZnO:Cu and PETN-ZnO:Fe composites was 26 and 95 J/cm^2^, respectively. The threshold duration of laser exposure before PETN-ZnO:Cu initiation was 50–60 ms, which made it potentially applicable in industrial photoinitiation systems. The possibility of initiating a PETN-ZnO:Cu composite by an infrared laser diode (11 W) with a wavelength of 808 nm, corresponding to the intrinsic absorption edge of CuO, was shown.

A joint analysis of the laser initiation thresholds of composites, the optical properties of the studied nanomaterials, and their comparison with literature data allowed us to justifiably assume the significant role of photochemical mechanisms in the initiation process of PETN-ZnO:Cu and PETN-ZnO:Fe composites. We concluded that the doping of zinc oxide nanopowder with impurity atoms of other metals or phases of narrower gap semiconductors can be an effective tool for tuning the ZnO photocatalytic activity for the initiation of energetic materials via visible lasers. The obtained results create the opportunity of developing industrial laser initiation systems based on common and affordable laser diodes and industrial energy materials, such as PETN.

## Figures and Tables

**Figure 1 nanomaterials-12-04176-f001:**
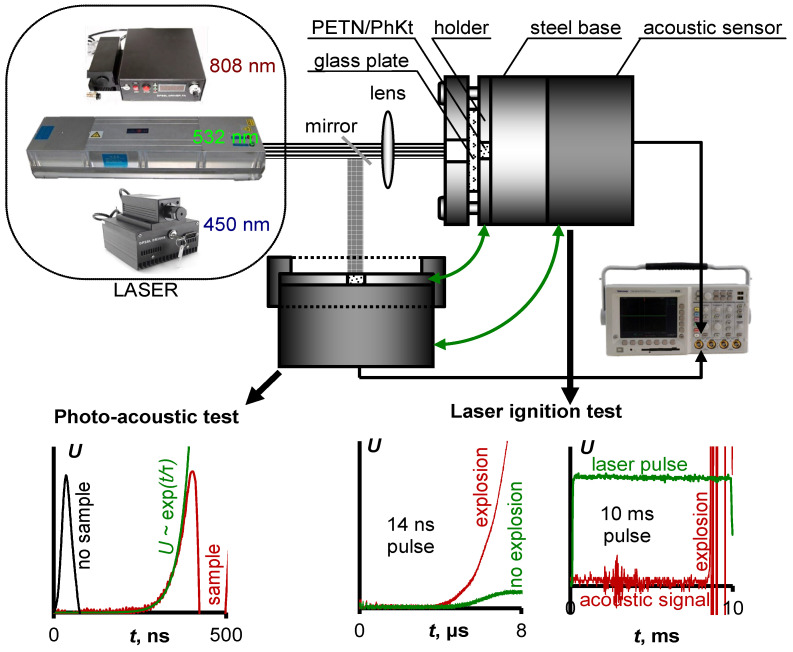
The schematic diagram of photoaucoustic measurements and laser initiation tests.

**Figure 2 nanomaterials-12-04176-f002:**
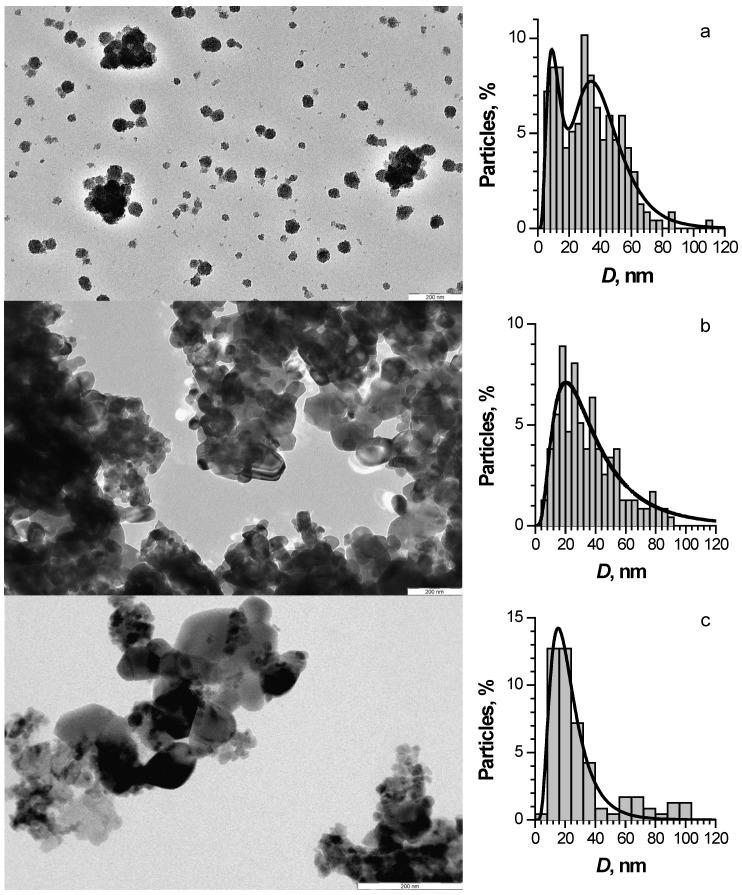
TEM images and particle size distributions of pure ZnO (**a**), ZnO:Cu (**b**), and ZnO:Fe (**c**) nanopowders.

**Figure 3 nanomaterials-12-04176-f003:**
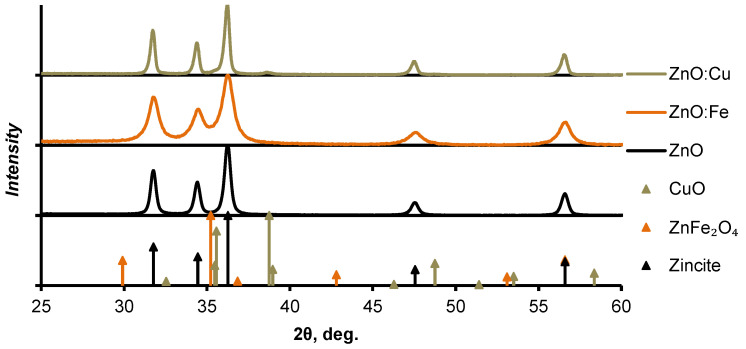
X-ray diffractograms of the pure ZnO, ZnO:Cu, and ZnO:Fe nanopowders. XRD patterns of the zincite [39], copper (II) oxide [38], and zinc ferrite [40].

**Figure 4 nanomaterials-12-04176-f004:**
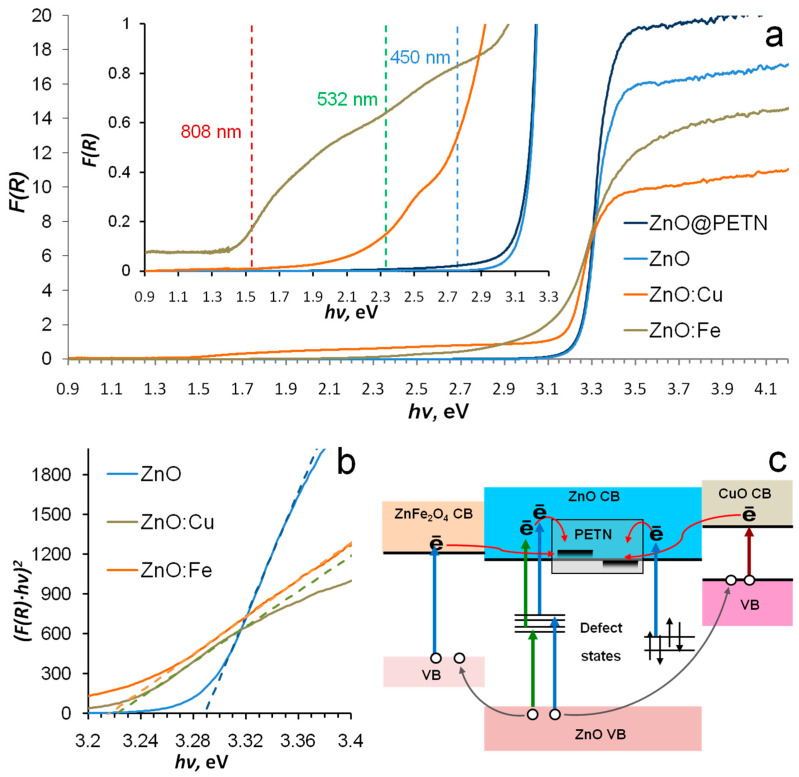
Diffuse reflection spectrum of the pure ZnO, ZnO:Cu, and ZnO:Fe nanopowders (**a**,**b**). Proposed mechanism of photoinduced charge transfer in ZnO:Fe−PETN and ZnO:Cu−PETN composites (**c**).

**Figure 5 nanomaterials-12-04176-f005:**
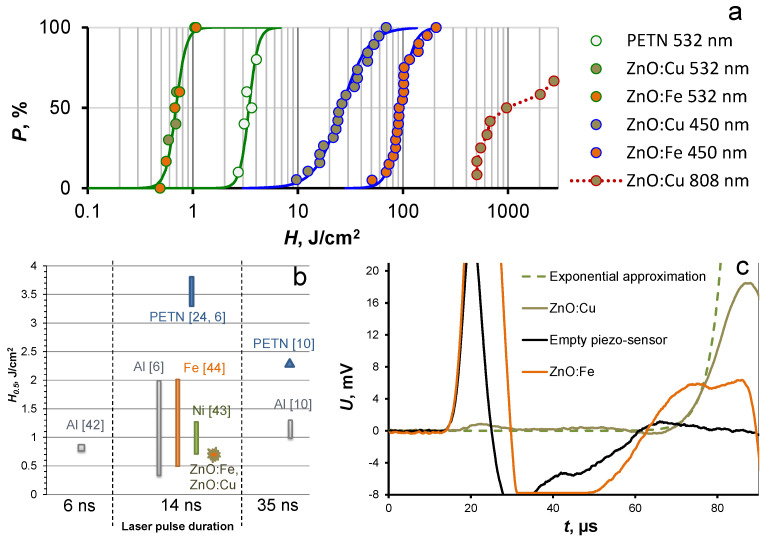
Explosion probability curves of the pure PETN, PETN−ZnO:Fe, and PETN−ZnO:Cu composites initiated by second harmonic of the YAG:Nd laser (532 nm), CW laser diodes with wavelength 450 and 808 nm (**a**). Comparison of the threshold energy density of the PETN−ZnO:Cu and ZnO:Fe (532 nm) and PETN (532 and 1064 nm) composites with iron (Fe; 1064 nm), nickel (Ni; 1064 nm), and aluminum (Al; 1064 and 532 nm) nanopowders initiated by nanosecond pulsed laser radiation (**b**). Oscillograms of photoacoustic response from the PETN−ZnO:Fe and PETN−ZnO:Cu composites (**c**).

**Figure 6 nanomaterials-12-04176-f006:**
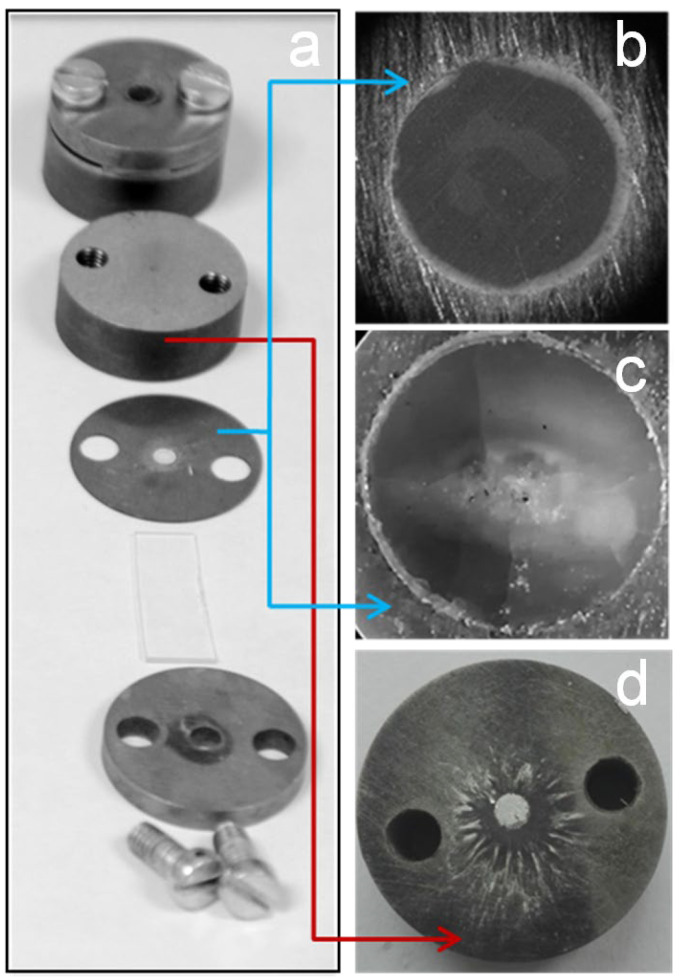
Photograph of the experimental setup for initiation tests (**a**), PETN-ZnO:Cu sample before (**b**) and after (**c**) laser irradiation. Aluminum foil on the steel base after the sample explosion (**d**).

**Table 1 nanomaterials-12-04176-t001:** Surface textural characteristics of the obtained nanopowders.

Sample	S_BET_, m^2^/g	V_Σ_, cm^3^/g	V_micro_, cm^3^/g	V_meso_, cm^3^/g	D_pores_, Å
ZnO	17.0	0.0876	0.0005	0.0860	202
ZnO:Fe	30.0	0.1455	0.0015	0.1430	191
ZnO:Cu	9.5	0.0319	0.0003	0.0299	126

## Data Availability

Additional data is available upon request to the corresponding authors.

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
