# Peer review of "Iron and Copper Doped Zinc Oxide Nanopowders as a Sensitizer of Industrial Energetic Materials to Visible Laser Radiation"

_nanomaterials, 2022, doi:10.3390/nano12234176_

Round 1
Reviewer 1 Report
The manuscript titled: "Iron and copper doped zinc oxide nanopowders as a sensitizer of industrial energetic materials to visible laser radiation" by Zverev et al. presents an experimental demonstration of a technique of applying the common industrial high explosive pentaerythritol tetranitrate as a photosensitive energetic material by adding zinc oxide nanopowders doped by copper and iron. The obtained energetic composites can be reliably initiated by a CW blue laser diode with a wavelength of 450 nm and power of 21 W. In addition, it is shown that ZnO:Cu composites can be initiated by an infrared laser diode with a wavelength of 808 nm. Based on the obtained results Authors porpose using nanomaterials based on zinc oxide as a sensitizer of industrial energetic materials to visible laser radiation. The manuscript well written and organized. The results are scintifically sound. Therefore, the manuscript is recomeded for publication in Nanomaterials.
Author Response
The authors are grateful for the review
Reviewer 2 Report
Zverev et al. introduced a technique using zinc oxide nano-powders (doped by copper and iron) spiked PETN as a photo-sensitive energetic material for laser-induced explosive reaction. The manuscript is well-written and well-organized, and I have only minor concerns:
1. I understand the theory of doping Fe and Cu to ZnO but is there a specific reason to use 5% molar ratios of Fe and Cu to Zinc oxide? Have the authors tried other molar ratios?
2. For the CW laser initiation experiment, the authors have mentioned in section 2.4 that the laser diodes have a Gaussian distribution which is very true. Since energy density is a ballpark estimate, in this case, have authors ever tested if the intensity distribution can affect laser initiation results as different laser diodes or diode modules that use other optics could have different gaussian characteristics?
3. Continued on the previous point, can the authors provide the characteristic beam size (FWHM) or simply photos of the two diode laser beams in the supplemental?
4. Line 299-303, Can the authors rewrite this part to make it more clear why an extremely higher threshold energy density is required using a laser diode?
5. Figure 5b, what laser wavelengths were the PETN composites' threshold energy density taken at?
6. Typo: Line 128 Photoacoustic
Author Response
- I understand the theory of doping Fe and Cu to ZnO but is there a specific reason to use 5% molar ratios of Fe and Cu to Zinc oxide? Have the authors tried other molar ratios?
We see the demonstration of the ability to apply obtained nanocomposite materials and a photochemical approach for the laser initiation of energetic materials as a critical goal of this study. We think that considering the two obtained materials is sufficient for this purpose. Varying the copper or iron concentration would significantly increase experimental work and article volume. But we agree that tuning the nanomaterial composition, morphology, and its content in EM are instruments for the development of optimal photosensitive energetic composites.
- For the CW laser initiation experiment, the authors have mentioned in section 2.4 that the laser diodes have a Gaussian distribution which is very true. Since energy density is a ballpark estimate, in this case, have authors ever tested if the intensity distribution can affect laser initiation results as different laser diodes or diode modules that use other optics could have different gaussian characteristics?
We didn't test who distribution of laser intensity affects thresholds. Energy distribution obviously can affect on results of the tests. We suppose that the high scattering index of our samples can offset this effect because it makes energy density inside the sample becomes more uniform than on the surface. The experimental condition for all samples, including pure PETN, as well as the experimental conditions of our previous work, which we used, stay constant, allowing us to compare results and estimate the efficiency of each photosensitizer.
- Continued on the previous point, can the authors provide the characteristic beam size (FWHM) or simply photos of the two diode laser beams in the supplemental?
Unfortunately, we cannot provide photographs of beam profile prints for Nd:YAG and 450 nm diode laser since they are now used in other experimental setups. We added supplementary materials that consist of energy distribution across a laser beam of Nd:YAG and 450 nm diode laser measured by a set of diaphragms with different diameters. These measurements were made during experiments described in the article. Supplementary materials also contain the energy distribution for the 808 nm diode estimated using a beam print. We also added the main text of the article.
- Line 299-303, Can the authors rewrite this part to make it more clear why an extremely higher threshold energy density is required using a laser diode?
We mean that CW thresholds seem to be extremely high only compared to the threshold energy density of pulsed laser tests. We agree that this idea was not presented clearly enough. We rewrote this part.
- Figure 5b, what laser wavelengths were the PETN composites' threshold energy density taken at?
Both fundamental frequency (1064 nm, 1.17 eV) and second harmonic (532 nm, 2.33 eV) was used in these tests. Since only the thermal mechanism of initiation is considered for metal nanoparticles, it seems to us that the use of these data is relevant. We added the information about initiation wavelength to the caption of figure 5.
- Typo: Line 128 Photoacoustic
Corrected